# The Social Dimensions of an Incentive-Based Urban Recycling Program: A Case-Study from Istanbul, Turkey

Betul Hande Gursoy Haksevenler [1], Aydin Akpinar [2] and Hiroshan Hettiarachchi [3],*

1   Department of Political Science and Public Administration, Faculty of Political Science, Marmara University, Kadıköy, Istanbul 34722, Türkiye; hande.gursoy@marmara.edu.tr
2   Department of Local Governments, Faculty of Political Science, Marmara University, Kadıköy, Istanbul 34722, Türkiye; aydin.akpinar@marmara.edu.tr
3   Independent Researcher, 650 Trailwood Path, Unit D, Bloomfield Hills, MI 48301, USA
*   Correspondence: hiroshanh@gmail.com

**Abstract:** Incentive-based programs are increasingly becoming common in recycling promotion. These programs are usually designed on the premise that the incentives will nudge the participants' behavior to be more pro-recycling so that they may continue to support recycling even if there is no reward presented anymore. The technical and economic aspects receive a great deal of attention, while the social elements do not usually receive the same level of attention as above. In this study, a survey was conducted to recognize the recycling consciousness of participants in an urban recycling promotional program. The urban recycling program analyzed in this study was administered by Zeytinburnu Municipality in Istanbul, Turkey. This program had about 10,000 active users at the time of this face-to-face survey conducted among 428 participants in 2021. Zeytinburnu is a densely populated municipality, with a greater majority residing in apartment buildings. The results revealed that about half of the survey population was spontaneous recyclers, and they would like to continue even if there was no reward, while the other half was in the program just for the reward. However, the findings also displayed the potential of the incentive mechanism to be used to enhance the pro-recycling behavior of its participants. This may be achieved through educational tools and providing solutions to ease the burden most apartment-dwelling participants carry with storage/transportation issues.

**Keywords:** recycling; source separation; waste collection; incentives; recycle consciousness; waste management in municipalities





## 1. Introduction

The global waste generation is rising as a result of the increasing population, urbanization, and industrialization, as well as per capita consumption. Proper management of waste is essential, as its mismanagement has a detrimental impact on public health and the environment [1,2]. Since this is an aspect that is particularly important to large population centers, municipal solid waste (MSW) management has become a significant indicator in measuring urban sustainability and the quality of urban living [2–5].

Modern day waste management strategies have incorporated many sustainable options, such as reduce, reuse, and recover/recycle (i.e., RRR concept), to minimize the volume of the waste stream that requires a final disposal. While reduce/reuse options take time to bring results, as instilling such values in a society requires substantial time, recovery/recycle is a practical solution that can be implemented relatively quickly. Perhaps this is why incentive schemes are one of the most commonly used practices. Oh and Hettiarachchi (2020) argued that solutions involving active participation of people have the potential to be sustainable in any city/community irrespective of their economic development. However, the long-term success of such a program heavily depends on the growth of public support and their continual active involvement [6].



The economic incentives are often used by the policymakers as an effective tool to improve public participation in environmentally friendly choices and strengthen their sense of responsibility [7,8]. Thus, incentives such as pricing schemes, gifts, and rewards are being used in many countries in conjunction with recycling promotional programs [9]. The use of such incentive schemes in promoting recycling and evaluating their effectiveness are topics that are important to all involved stakeholders. Although there are a lot of discussions about the technical/economic aspects of recycling programs, the same is not true about the social dimensions of recycling (i.e., analyzing the participants and their behavior). The consideration of factors influencing public behavior is often overlooked during the planning/executing of recycling programs [10,11]. This difference was also noticeable in the corresponding academic literature. Although there has been some increase in recent years, the number of studies in the literature remains limited, with most studies focusing on certain specific geographical regions [10–13].

Analyzing the social dimensions is potentially the best way to find out if/why a recycling program works (or not) in a certain community, as such analysis may reveal information about participants' motivation to recycle, challenges they face while participating, and also their opinions on what is working and what is not working. These data may be used to make recycling programs more comfortable/fitting to the participants' environment, which, in return, can not only assist with retention, but also increase the membership. An incentive-based recycling program provides an excellent opportunity to push the public to think/act in a recycling-friendly manner. However, there is also a notable concern about a drop in motivation once the financial incentives are withdrawn and that the desired results may not be achieved [14,15]. Therefore, it is important to nudge people for a behavioral change, to make them ultimately feel responsible for recycling, even if the incentives cease to exist at one point. In general, being pro-recycling (or recycling conscious) can be characterized as a branch of the pro-environmental behavior, as the individuals with pro-environmental behavior consciously choose to minimize the negative impact of their actions on the environment [16].

In this context, the objective of this study was to present an investigation conducted to analyze the social dimensions of an urban recycling program. The case analyzed here was one of the municipalities in Istanbul, Turkey, called Zeytinburnu. The research objective was to recognize the pro-recycling consciousness among the program participants and relate it to the demographics and other inputs collected through a survey. The data were collected in 2021 through face-to-face interviews with out of ~10,000 active participants within Zeytinburnu's urban recycling promotion program. Following a brief background discussion in the next section covering the latest relevant findings from the literature, the subsequent sections present details of the case study, the survey methodology employed, the results, and the analysis.

## 2. Background

Recycling through source separation of MSW (i.e., separation of waste at the point of generation) only works when the waste generator is able to comply. Those who lack the necessary knowledge and understanding may not be able to contribute [17]. For instance, a survey conducted by Keramitsoglou and Tsagarakis (2013) observed a high emphasis on e-waste recycling (such as batteries, fluorescent and energy-saving lamps, and other electrical items) when the participants were knowledgeable about the adverse environmental impact e-waste can make [18].

However, having such knowledge does not necessarily translate into action. Many individuals express support for recycling, but often fail to take appropriate measures due to various other reasons, such as limited space (to store recyclable wastes), lack of time, household attitudes, apathy, insufficient motivation, and inadequate collection services [19]. Since the voluntary recycling efforts of individuals do not always reach the desired level, regulations that involve rewards/punishments are often used to increase participation [12]. Previous research has found the reward-based practices to be more effective and receptive

to the public than any punishment-based schemes in general [20,21]; however, monetary incentives may lead to both positive and negative outcomes [22]. One example of an economic incentive is to give out coupons (based on the amount of waste they recycle) that the participants can use at local markets [23]. This approach has been successful in a few major cities like London and Philadelphia, as well as in some parts of Latin America [21,24,25]. In Mexico City, residents exchange recyclables for "puntos verdes" (green dots) that can be used at local farmers' markets to buy fresh produce [26].

Instilling proper waste-sorting habits in public is particularly important. Properly designed incentives could encourage the public to adhere to desired laws and regulations at a low cost to society [27]. Numerous previous studies have examined the impact of economic incentives on waste management [8,28–31]. The general consensus is that the behavioral changes resulting from economic incentives are more likely to be sustained compared to interventions based solely on knowledge [22]. Although the financial incentives have been thought to make a positive effect on waste recycling behaviors in general [28,32], a few studies have expressed doubts about their effectiveness [33–35].

A study conducted by Xu et al. (2018) on the effect of economic incentives and social impact strategies on waste separation practices revealed that the former was more effective than the latter. The same study also reported that economic instruments may only efficiently work at an early stage of promoting waste separation, but the effects of the social norms built during this stage may be long lasting [36]. Truelove et al. (2014) stated that monetary incentives could create negative consequences in the long run, as they prevent environmental tendencies in general, such as environmental self-identity and/or personal ecological norms [37]. It is also worth mentioning that making recycling less costly/effortful may also be effective in promoting its adoption [15,38].

In Turkey, the importance of waste management was realized relatively later. Therefore, recycling and recovery are topics that have yet to receive popularity within this country. As a result, the number of studies conducted on implementation of recycling/recovery activities in Turkey is limited. This limited literature covering such programs in Turkey has mainly focused on the technical dimensions (such as Yalcinkaya and Uzer, 2022 [39]), while only a very few studies discussed the social dimensions [40,41]. As per the literature, no studies have been conducted yet on recycling programs initiated by any of the Turkish municipalities: the case presented and analyzed is probably the first such study.

## 3. Case Study: Zeytinburnu Municipality, Istanbul, Turkey

About 32.3 million tons of MSW is generated annually in Turkey (Turkish Statistical Institute 2021). The majority of this MSW is being disposed of at landfills, while a growing (but still minor) fraction is being recycled. In 2017, the Ministry of Environment and Urbanization predicted the recycling rate to reach 35% by 2023 [42], but it was still at 12% by 2021, while the rest of the MSW was disposed in regular or unsanitary landfills [43]. In 2019, Turkey introduced the Turkish Zero Waste Regulation (ZWR), which was perhaps the most important waste management-related step the country has taken in recent times [44]. The ZWR not only promotes recycling, but also encourages waste elimination/minimization through changing consumption habits. The ZWR requires each municipality with a population greater than 250,000 to establish their own waste recycling programs.

### 3.1. Zeytinburnu Municipality, Istanbul, Turkey

The city of Istanbul comprises one metropolitan municipality and 39 district municipalities. Ideally, all of these municipalities are obligated to facilitate recycling within their districts as per the ZWR requirement, as the population in each of these municipalities is over 250,000. However, only a few of them have taken steps to fulfill this responsibility thus far. Zeytinburnu, the municipality selected for this study, is one of the few that has complied with the ZWR by launching a waste separation initiative called the "Waste Recovery Project". The relative location of Zeytinburnu Municipality within Istanbul is illustrated in Figure 1.

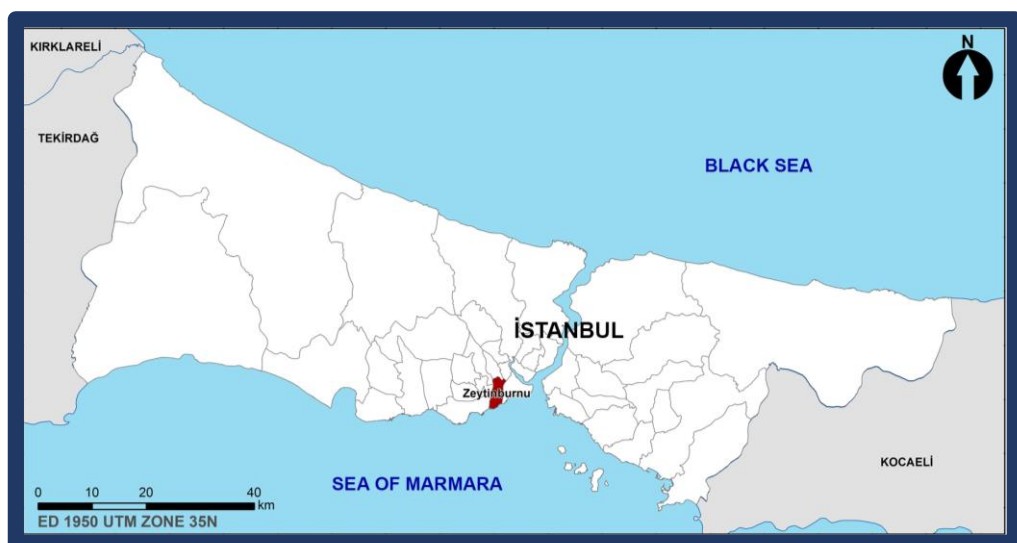

**Figure 1.** Location of Zeytinburnu Municipality within Istanbul.

The Zeytinburnu district has about 293,000 inhabitants living in 11.40 km$^2$, making it one of the densest populations (25,700 people/km$^2$) in the Istanbul metro area [45]. As per the data published by the Turkish Statistical Institute (2022), the population of Zeytinburnu is relatively young in age: only 3.4% of them are older than 65, while 38.3% are under the age of 24. According to the same source of data, both the income and the education of the residents are not high. The average household income in the district is well below the poverty line in Turkey, which is USD 1675/month for a family of four. In terms of literacy, 4.3% of residents were illiterate, 31.2% were literate up to primary school level, 24.6% secondary school, 23.7% were high school graduates, and only 15.2% were university graduates [46].

Before launching Zeytinburnu's Waste Recovery Project, mixed waste was collected daily from each street by the municipality. As the new project is still limited by its capacity and coverage area, the municipality continues to collect mixed waste on each street. A waste collection fee is included in their water utility bills (6.3 cents/m$^3$ of water).

### 3.2. Zeytinburnu Waste Recovery Project

Under Zeytinburnu's Waste Recovery Project, the residents of the municipality district are able to participate in recycling and get rewarded for their contribution. Their contribution involves bringing pre-sorted recyclable waste to designated collection centers, which are either permanent centers (fixed units) or the collection vehicles that come to certain neighborhoods on certain days (mobile units). A few such collection centers are shown in Figure 2.

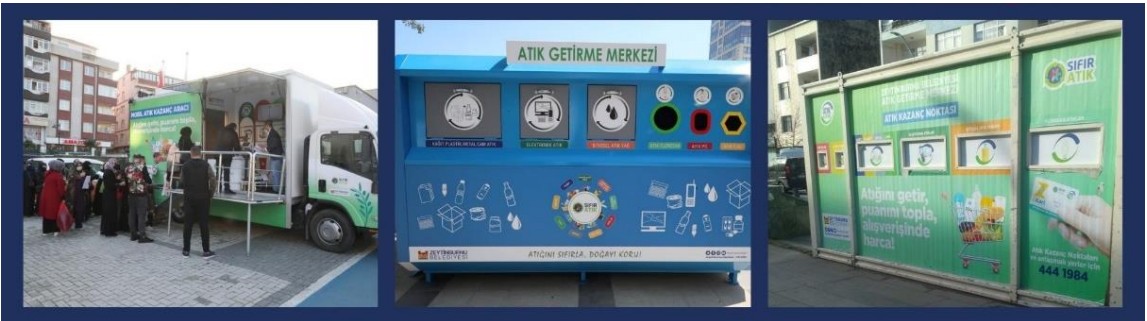

**Figure 2.** Recyclable collection centers in Zeytinburnu Municipality: a mobile collection center with staff (**left**) and two automated collection facilities at fixed points (**middle** and **right**).

Waste sorted into paper, plastic, metal, glass, used cooking oils, batteries, fluorescent bulbs, and other electronics are collected via mobile units or collection centers located in six neighborhoods. Incorrectly sorted recyclables are not accepted. Residents receive a certain amount of money for the waste they bring in, but not in cash. Instead, the payments received are loaded onto a municipality-issued Z-card (i.e., a plastic card that works like a gift card or a debit card). The balance in the Z-card can be used towards buying things at the local markets. The payments are made according to the type and weight of the waste. For example, 1 kg of packaging waste or paper receives USD 7 cents, and for plastics it is USD 10 cents. Used cooking oil receives USD 20 cents per 1 kg. Electronic wastes, batteries, and fluorescent bulbs receive USD 16, 40, and 2 cents, respectively. This Z-card is only used by the Zeytinburnu residents, and only for this recycling program. Based on the Z-card registration data (Zeytinburnu Municipality IT Directorate), about 10,000 residents participated in this recycling program in 2021.

## 4. Research Methodology

A survey consisting of a questionnaire was employed to collect data in this study. This survey was conducted in December of 2021 at the collection centers that change location every half a day. The target survey population consisted of Z-card-holding residents of the district. In 2021, there were approximately 10,000 active Z-cards users. The minimum sample size required for the survey was determined using the Raosoft (Version 2007) calculator, a software tool designed for sample size calculation [47]. The minimum number of participants required for the survey was calculated to be 370 people, with a 5% margin of error and a 95% confidence interval. At the end, 428 Z-card holders were interviewed for the survey at mobile/fixed waste collection centers (see Figure 3). Participation in this survey was on a voluntary basis. The purpose of this study and this process were explained to each participant before starting the survey. During this survey, the participants were interviewed face to face, and the responses were manually recorded on paper.

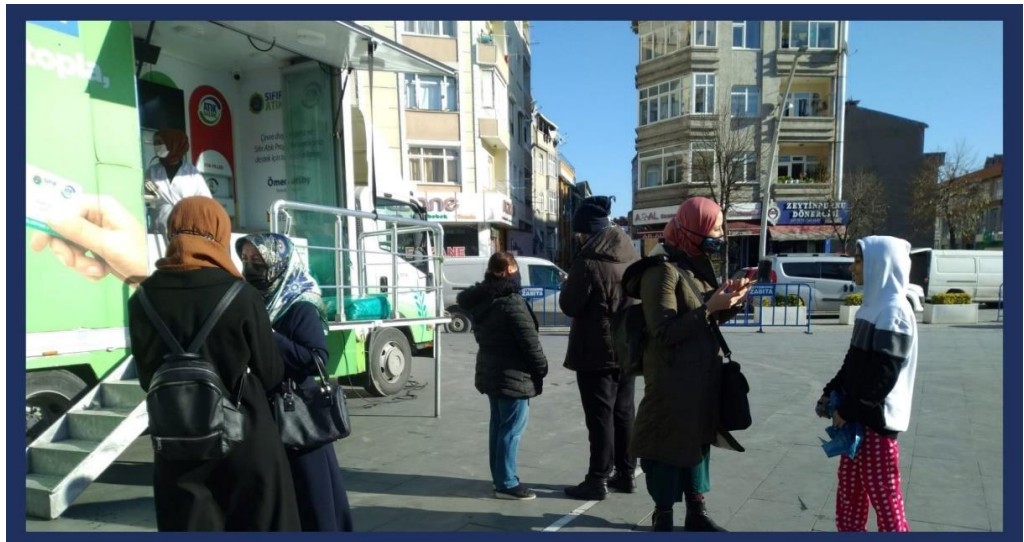

**Figure 3.** A survey being conducted at a mobile recyclable collection center in Zeytinburnu Municipality.

### 4.1. Survey Design

This survey included two parts: The first part for demographic information of the survey takers and the second part to collect information on their knowledge/thinking/opinions about the ongoing recycling program implemented by Zeytinburnu Municipality. The demographic questions in the first part of the survey were designed to collect information on gender, age, marital status, education, occupation, and the number of people living in the same household.

The questions in the second part of this survey were aimed at assessing participants' level of awareness, knowledge, responsibilities, ideas for increasing efficiency, and the difficulties they faced in waste sorting. The first question (Q1: Which of the following do you think is non-recyclable waste?) was asked to assess the participants' general knowledge of recycling. The second question (Q2: How did you learn about the Zero Waste Regulation) was asked to understand how participants accessed recycling-related information. The third question (Q3: Who do you think should implement the separate waste collection?) was designed to evaluate to what extent the participants feel responsible for recycling. The next question was about the obstacles the participants were facing while participating in the program (Q4: What is the major challenge to carry waste to the collection/storage area?). The objective of the fifth question was to understand how recycling could be made more active and effective in the eyes of the participants (Q5: What should be done to make separate waste collection more active and effective?). Perhaps the most important question is the last one (Q6: Would you still separate your waste even if there was no incentive given?), which was included to judge participants' interest in continuing to recycle without receiving any rewards.

### 4.2. Demographics of the Survey Population

The demographic data collected are summarized in Table 1. A few dominant groups were observed among the survey takers: 82% of the participants were women, 54% were between the ages of 41–65, 70% were married with children, 75% were primary school graduates, 61% were housewives, and 45% of them lived with 3–5 others in the same household.

**Table 1.** Profile of respondents participating in the survey.

| Demographic Information | Number of Responses | Percentage (%) |
|---|---|---|
| **Gender** | | |
| *Male* | 77 | 18 |
| *Female* | 351 | 82 |
| **Age** | | |
| *Under 18* | 53 | 12 |
| *18–24* | 19 | 4 |
| *25–40* | 114 | 27 |
| *41–65* | 231 | 54 |
| *Over 65* | 11 | 3 |
| **Marital status** | | |
| *Single* | 123 | 29 |
| *Married/with child* | 298 | 70 |
| *Married/without child* | 7 | 1 |
| **Education** | | |
| *Primary school* | 319 | 74 |
| *High school* | 72 | 17 |
| *Tertiary school* | 37 | 9 |
| **Occupation** | | |
| *Housewife* | 261 | 61 |
| *White collar* | 24 | 6 |
| *Blue collar* | 55 | 13 |
| **Number of people living in the same household** | | |
| *Under 3* | 128 | 30 |
| *3–5* | 192 | 45 |
| *Over 5* | 108 | 25 |

Having a very high participation level of women in this survey was an interesting observation. While we are not able to rule out the possibility of women being more active in recycling, this high percentage of women participants in our data was most likely to be associated with other reasons, such as the time and location of the survey. As mentioned

before, this survey was conducted near the recycling collection centers (see Figure 3), and the survey interviews took place during the daytime when the majority of the Zeytinburnu male population is at work. It may be also related to the fact that, following the local culture, Zeytinburnu women have a relatively high responsibility in the kitchen.

## 5. Results and Discussion

The six questions posed to the survey participants and the multiple-choice answers made available for each question are summarized in Table 2 (column 1). These six questions were designed to collect information covering three general aspects pertinent to this discussion: Q1 and Q2 to assess participants' knowledge/understanding of recycling/the ZWR; Q3, Q4, and Q5 to collect participants' opinions on source-separated waste collection; and finally, Q6 to assess the pro-recycling behavior of the survey participants. The number of responses received for each answer and the same reported as percentages are also presented in Table 2 (columns 2 and 3, respectively) and the same results are graphically presented using bar charts in Figure 4. Each of these aspects has been discussed briefly in the next few subsections.

**Table 2.** A summary of the responses received for the six survey questions.

| What Is Expected to Be Assessed with This Question Is Given in Parentheses | Number of Responses | Percentage (%) |
|---|---|---|
| **1. Which of the following do you think is non-recyclable waste? (participant's common sense in recycling)** | | |
| *Diaper* | 295 | 68.93% |
| *Coke can* | 18 | 4.21% |
| *Battery* | 7 | 1.64% |
| *Fluorescent Lamp* | 2 | 0.47% |
| *All of them* | 106 | 24.77% |
| **2. How did you learn about the Zero Waste Regulation? (means of access to recycling information)** | | |
| *Application examples of municipality* | 288 | 67.29 |
| *Friend/neighbor/relative* | 76 | 17.76 |
| *Television/public spotlight* | 31 | 7.24 |
| *Internet/social media* | 28 | 6.54 |
| *Billboard* | 4 | 0.93 |
| *Official gazette/regulations* | 1 | 0.23 |
| **3. Who do you think should implement the separate waste collection? (opinion on who needs to be responsible for recycling)** | | |
| *Municipalities should* | 268 | 62.62 |
| *Each individual should* | 103 | 24.07 |
| *Presidency/ministry should* | 44 | 10.28 |
| *Governors should* | 10 | 2.34 |
| *There is no need for waste management* | 2 | 0.47 |
| *The private sector should* | 1 | 0.23 |
| **4. What is the major challenge to carry waste to the collection/storage area? (difficulties they may encounter in participating)** | | |
| *Transporting waste to the collection/storage area* | 196 | 45.79 |
| *Insufficient space to separate waste* | 137 | 32.01 |
| *The lack of awareness sharing the same place* | 46 | 10.75 |
| *Insufficient time to separate waste* | 34 | 7.94 |
| *Lack of knowledge about how to separate waste* | 15 | 3.50 |
| **5. What should be done to make separate waste collection more active and effective? (how to make recycling more active and effective)** | | |
| *Basic education should be given* | 186 | 43.46 |
| *Incentives should be given* | 146 | 34.11 |
| *Attention-grabbing directions should be made* | 66 | 15.42 |
| *Penalties should be imposed* | 30 | 7.01 |
| **6. Would you still separate your waste even if there was no incentive given? (the relationship between the incentives versus pro-recycling behavior)** | | |
| *Yes, I would* | 218 | 50.93 |
| *No, I would not* | 210 | 49.07 |

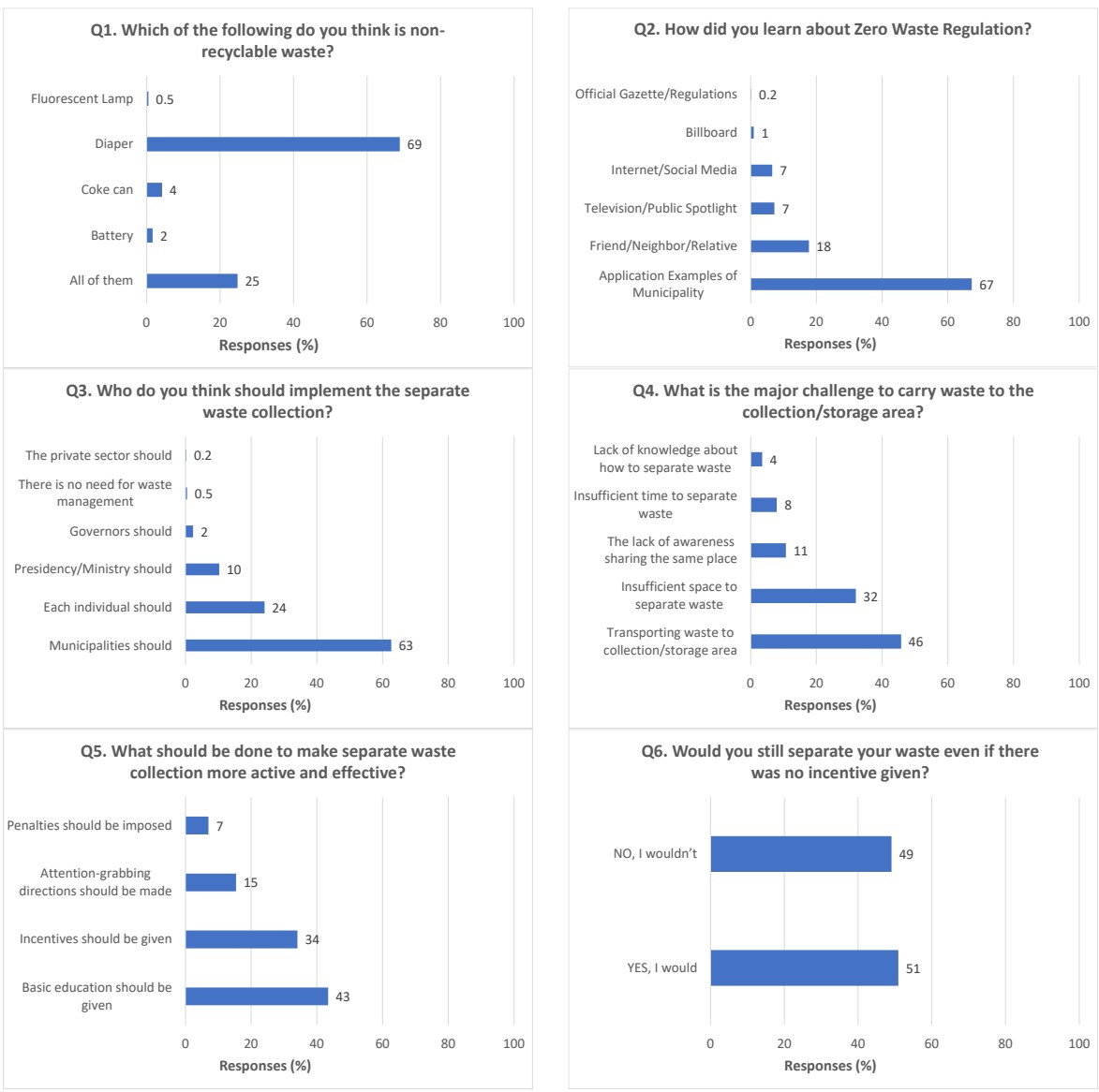

**Figure 4.** Graphical representation of the responses received for the six survey questions.

*5.1. Participants' Knowledge/Understanding of Recycling/the ZWR*

Knowledge being a prerequisite to action is a well-established fact [48]. This is also quite true for recycling, and there is sufficient evidence in the literature to show how knowledge and information play an important role in increasing participation in recycling programs [49–51]. Following waste recycling classifications in source separation can be a challenging task for many [52]. For example, multi-component packaging, such as Tetra Pak (e.g., milk and juice boxes), can create confusion about what category they belong to [13].

In this context, the answers given to the first two questions provide some important information about the waste management literacy of the survey takers. The first question (Q1: Which of the following do you think is non-recyclable waste?) was followed by a few multiple-choice answers, but there was only one correct answer (i.e., diaper). A little over two-thirds (~69%) of the participants getting it right indicated that this survey population was quite literate in waste sorting and recyclability of different waste types (Table 2). Although soda cans and water bottles are often considered as symbols of recycling and frequently used in recycling educational/promotional material, about 4% of the survey takers still did not recognize soda cans to be recyclable. On the other hand, it was encour-

aging to observe that this fraction, which did not recognize the recyclability of batteries and fluorescent bulbs, was limited to about 2% of the survey population. The major surprise in this category was those who selected "all of the above" as their response, which was 25%. It is possible that uncommon recyclable items, such as batteries and fluorescent bulbs, may have indirectly forced some respondents to select "all of the above" as the correct answer. Recycling electronics, such as batteries and fluorescent bulbs, has not become commonplace yet in many countries/cities, as most municipalities do not offer such services.

In Q2, participants were asked how they found out about the ZWR. Interestingly, two-thirds of them (~67%) acquired this prior knowledge thanks to the waste recovery project launched by their own municipality (Table 2). Word of mouth (~18%) and digital media (~14%) also significantly contributed to the spread of the message. It was also interesting to notice that the traditional means such as billboards and governmental gazettes (slightly over 1%) may not be the best tools to rely upon. Overall, about 85% learned about the recycling program by paying attention to their surroundings, such as seeing the recycling program being conducted by the municipality (67%) and listening to people around them (~18%).

*5.2. Participants' Opinions/Experience Related to Separate Waste Collection*

Traditionally, MSW management has been a governmental business: while the relevant government authority makes the rules/regulations, the local municipalities (or their designees) take care of collection and disposal. This has probably given rise to the misconception that the local municipality should continue to take the sole responsibility of the process, despite the fact that modern-day waste management is a rather complex process that needs some contribution from all stakeholders, especially the waste producers. This thinking was also exhibited in the responses received for Q3: Who do you think should implement the recycling? About 63% of the participants said that it should be conducted by the municipalities, while 24% and 10% felt that the responsibility must go to the individuals and the government, respectively (Table 2).

The municipalities taking sole responsibility may have been a sensible choice in the past, when waste management merely meant collection and disposal. However, this process has now evolved to be more sustainable to include better options, such as recycling/recovery that requires the waste producers to take some responsibility as well. In fact, source separation of waste is now being recognized as the first step towards sustainable waste management [25,27]. However, many are not ready to part ways with their traditional thinking/experience. In other words, there is a perception among some that the responsibility for separation and recycling at the source should lie within the municipalities rather than individuals [53]. One of the arguments people often use to justify this thinking is the taxes that they already pay to the government/municipality to obtain such services [54].

The fact that three-quarters of the population are not willing to recognize any individual responsibility for recycling is certainly a challenging proposition. Perhaps, Zeytinburnu Municipality should consider educational/awareness-raising avenues to close this gap. In order to develop a sense of responsibility for recycling, it is necessary for the citizens to understand that their own behavior can make a significant impact on how their waste is being managed [55]. The general hypothesis is that those who feel responsible for the waste generation tend to separate their waste [14].

When asked about the possible difficulties that could prevent recycling (Q4), 46% of the participants ranked transportation of the source-separated material to collection centers, while 32% stated not having enough space at home to separate the waste, as the biggest challenge. However, it was somewhat encouraging to see that only a few respondents mentioned time/knowledge as issues. "Insufficient time to separate waste" and "Lack of knowledge about how to separate waste" only received 8% and 3.5%, respectively. This proved that the key issues are more related to a lack of infrastructure than a lack of will. Similar observations have been reported by previous studies. For example, Timlett and Williams (2011) discussed the importance of factors such as property status and type of

residence in making recycling more effective. They also emphasized on the necessity to place separate curbside collection systems for recyclable materials and organic waste to make source separation, especially at housing types such as apartments, more effective [56]. In another similar survey conducted by Sidique et al. (2010), the participants alluded to placing material in wrong bins (possibly due to a lack of knowledge) being a major issue. In their study, they also emphasized on the importance of making recycling-related information easily accessible to the public [57]. In a review study conducted by Knickmeyer (2020), the lack of necessary infrastructure was identified as one of the most important obstacles faced by recycling program participants. The strategic placement of recycle waste collection points (easy access in a short distance) and frequency of collection were identified as the features that could increase participation in recycling activities [11].

In order to collect participant opinion on how the recycling program may be made more active and effective, Q5 (What should be done to make recycling more active and effective?) was included in the survey. About 43% of the participants chose basic education, while 34% chose incentives, and about 15% chose having better instructions. Similar observations have been reported by other researchers from around the world, and many have argued that basic education and access to information play important roles in waste recycling [49–51]. The percentage of respondents who asked for more punitive actions were limited to just 7% in the current study. Data from the literature have also suggested that it is usually the case elsewhere. A study by Keramitsoglou and Tsagarakis (2013) concluded that, in general, the participants prefer policy measures that are based on rewards over punishments, or punishments that do not require personal responsibility [18].

*5.3. Pro-Recycling Behavior of the Survey Participants*

The last question in the survey was probably the one that is most useful to the decision makers at Zeytinburnu and/or other similar municipalities. This question (Q6 in Table 2) was designed to find out the pro-recycling inclination of our survey population by asking whether they would still recycle in the absence of a reward mechanism. Those who said "yes" were 51%, which is substantial. However, this also means that about half of the respondents (49%) were in the program just for the financial benefit. For the purpose of further analysis/discussion of this topic, we identified the respondents who answered "no" as the less recycle-conscious group (or less-conscious group, in short). In a similar way, those who said they would still separate without any reward were called the more recycle-conscious group (more-conscious group). Answers that were provided to Q1 (in Table 2) by the participants in both groups were cross-examined to see if there was any correlation between knowledge/understanding and their recycling consciousness. A slightly higher percentage in the more-conscious group had the correct answer to Q1 (i.e., diapers) compared to the less-conscious group: While 73% (159 out of 218 participants) of the more-conscious group had the correct answer, it was 65% (136 out of 210 participants) in the other group. A similar trend was noticed in the answers to Q3 (in Table 2): while 27% of the more-conscious group felt that the individuals should take the waste separation responsibility, those who felt the same way in the less-conscious group was 21%.

The above results indicate that the recycling program will lose half of its participants (i.e., the entire less-conscious group) if/when the incentives cease to exist. In order to have a self-running, long-lasting program, it is essential to substantially increase the size of the more-conscious group. What could Zeytinburnu Municipality do to increase the more-conscious group within its population? Although the above analysis does not provide a direct answer to this question, the same data could be analyzed further to uncover more trends. To obtain some insight, we looked at the demographic data in each group separately. What is presented in Table 3 is a side-by-side comparison of the demographic data for the more-conscious group versus the less-conscious group. In addition, the responses received from the more-conscious group are also graphically presented in Figure 5.

**Table 3.** Demographic breakdown of the more recycling-conscious group versus the less recycling-conscious group.

| Demographic Information | Responses from the More-Conscious Group | | Responses from the Less-Conscious Group | |
|---|---|---|---|---|
| | N | % | N | % |
| **Participants** | 218 | 50.93 | 210 | 49.07 |
| **Gender** | | | | |
| *Male* | 40 | 51.95 | 37 | 48.05 |
| *Female* | 178 | 50.71 | 173 | 49.07 |
| **Age** | | | | |
| *Under 18* | 26 | 49.06 | 27 | 50.94 |
| *18–24* | 11 | 57.89 | 8 | 42.11 |
| *25–40* | 67 | 58.77 | 47 | 41.23 |
| *41–65* | 109 | 47.19 | 122 | 52.81 |
| *Over 65* | 5 | 45.45 | 6 | 54.55 |
| **Marital status** | | | | |
| Single | 72 | 58.54 | 51 | 41.46 |
| Married/with child | 142 | 47.65 | 156 | 52.35 |
| Married/without child | 4 | 57.14 | 3 | 42.86 |
| **Education** | | | | |
| Primary school | 138 | 43.26 | 181 | 56.74 |
| High school | 50 | 69.44 | 22 | 30.56 |
| Tertiary school | 30 | 81.08 | 7 | 18.92 |
| **Occupation** | | | | |
| *Housewife* | 121 | 46.36 | 140 | 53.64 |
| *White collar* | 21 | 87.50 | 3 | 12.50 |
| *Blue collar* | 27 | 49.09 | 28 | 50.91 |
| *Retired* | 13 | 68.42 | 6 | 31.58 |
| *Others* | 36 | 52.17 | 33 | 47.83 |
| **Number of people living in the same household** | | | | |
| *Under 3* | 140 | 53.85 | 115 | 46.15 |
| *3–5* | 35 | 54.90 | 30 | 45.10 |
| *Over 5* | 43 | 39.81 | 65 | 60.19 |

The gender composition was approximately a 50/50 distribution within each group, which means that gender was not a decisive factor in the survey takers' recycling consciousness. However, age seemed to be an important indicator. Survey takers between the ages of 18 and 40 consistently showed more recycling consciousness compared to the other age groups. A vast majority of the survey participants were married and had children (70%, as seen in Table 1), but they were roughly equally divided between both groups (Table 3). The data presented in Table 3 also suggest that those who do not live with children (single or married/without children) show more recycling consciousness, which may be only justified by the time factor: they may be less busy and could allocate more time to engage in recycling compared to the parents/families with children.

It is natural to see increased recycling consciousness among those who have more knowledge of the topic. Education can be considered as an indicator that may be directly proportional to knowledge/exposure. As per the data presented in Table 1, a significant portion (74%) only had primary school education, while 17% had high school, and 9% had tertiary education. While there was no significant difference in the recycling consciousness among those who only had primary school education, in the other two categories with more education, a clear majority exhibited more recycling consciousness. Among the tertiary education category, those who showed more recycling consciousness were as high as 81%. Occupation of the survey participants could also work as an indirect indicator of their recycling knowledge/exposure based on the assumption that more education leads to a better job. The more-conscious proportion among white-collar jobs was significantly higher (87% versus 12%) compared to those who had blue-collar jobs (49% versus 51%). There was also a significantly high proportion of more recycling consciousness among

retirees, but understanding this trend is not possible without further information (such as retired from what kind of profession). Although our results exhibited some positive relationships between increased recycling consciousness with increased level of education, it is also important to mention that a few previous studies have suggested the effect of education/exposure on pro-recycling behavior to be less significant [18,19].

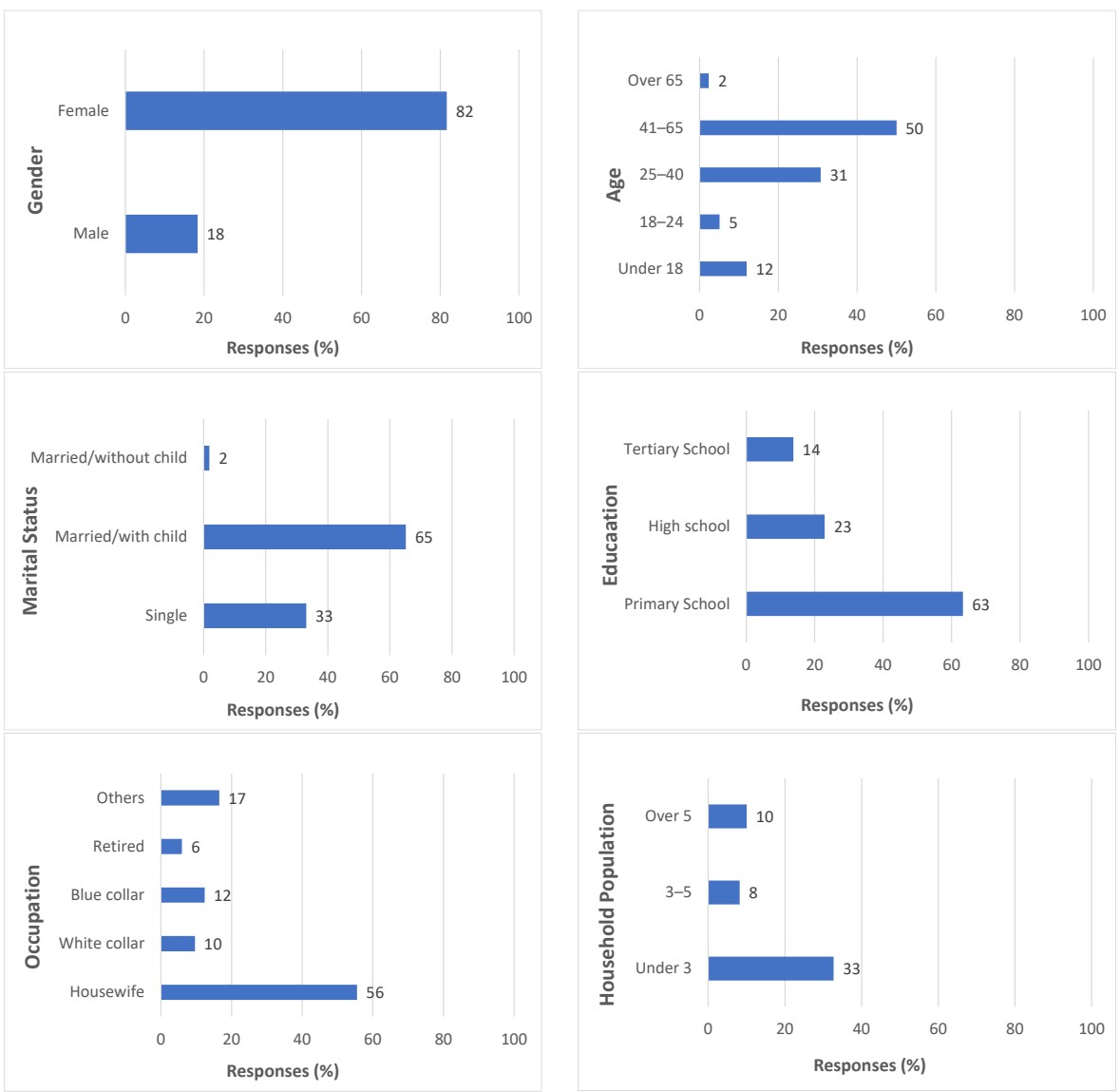

**Figure 5.** Graphical representation of the results in Table 3 for the more-conscious group.

Table 1 already suggests that 25% of the survey takers are from large families (>5 living in the same household). Their recycling-consciousness breakdown, shown in Table 3, indicated that the majority was less conscious (40% versus 60%). We believed that this could be due to the practical difficulty in maintaining proper waste management in a limited space: when the family is large, obviously there would be more waste to deal with too. During the survey, we also found out that almost all the survey takers (about 97%) live in apartment buildings (this aspect is not covered in Tables 1–3). Apartments are usually smaller than single houses, and most apartments are not equipped with extra space to manage waste, which could be the reason for this negative trend exhibited by the large families. Afterall, as mentioned before, Zeytinburnu is one of the most densely populated districts in Istanbul (25,700 people/km$^2$).

In order to compare the samples of the more-conscious and less-conscious groups, the one-way analysis of variance (ANOVA) test was used to determine whether the consciousness variable shows significance compared to other variables (Table 4). The results indicated the differences between the arithmetic means of the consciousness variable and the age, gender, marital status, household, occupation, and education variables to be significant. The differences in the arithmetic means between the consciousness variable and age, gender, and marital status were found not to be significant, since the null hypothesis could not be rejected. On the other hand, the differences in the arithmetic means between the consciousness variable and the household, occupation, and education variables were found to be significant after rejecting the null hypothesis. In other words, according to the results of the ANOVA test, the independent variables that were found to be related to consciousness are occupation, education, and household size.

**Table 4.** ANOVA test results showing the relationship between consciousness levels and demographic characterization of the groups.

| | F | Prob > F |
|---|---|---|
| Consciousness and gender | 0.04 | 0.8447 |
| Consciousness and education | 16.50 | 0.0000 * |
| Consciousness and marital status | 2.13 | 0.1207 |
| Consciousness and age | 1.17 | 0.3243 |
| Consciousness and household size | 3.62 | 0.0276 * |
| Consciousness and occupation | 4.50 | 0.0014 * |

Note: "*" indicates where the null hypothesis was rejected at the 5% significance level.

## 6. Lessons Learned

As mentioned before, the average income level in the Zeytinburnu district is below the poverty level [46]. This was also proven to be compatible with our survey population: purely on an observational basis, it appeared that the income levels of the survey participants were not high. This may be the reason for half of the survey population participating in the program just for the reward. While it is understandable to launch a recycling promotion program with economic incentives, a program that heavily depends on intensives is not a healthy mechanism for the long run. Ideally, an economic incentive-based promotional program should be a sunset program—a program that runs to achieve specific goals in a specific duration. If this is the case, Zeytinburnu's recycling promotional program must focus on instilling pro-recycling thinking among its population while it lasts.

Education is one aspect Zeytinburnu Municipality could capitalize on to transform its less recycle-conscious citizens into more-conscious ones. Interestingly, 43% of the survey takers also thought of better education as the primary factor that can make recycling efforts more active and effective, while incentives were ranked as the primary factor by only 34% (Table 2). Based on the results discussed above (especially the indicators related to the knowledge, education, and occupation), it was fair to conclude that those who have a better understanding/knowledge in waste management tend to be more serious and responsible about recycling (more conscious). While schools/colleges can be good breeding grounds for formal as well as informal awareness activities, the low interest shown by the older population (over the age of 40) suggests that these awareness-raising activities must somehow reach them as well. Community-based recycling awareness activities that also involve local businesses (such as grocery stores/supermarkets) and volunteer organizations might also help to make a difference.

Another important clue revealed by the survey results was about the practical difficulties the survey population is facing based on their lifestyle and/or housing arrangements. As mentioned before, almost all of them live in apartments, and have limited space to arrange/sort/store waste. In addition, as inferred by our data, parents/families with kids may not have the luxury to spend more time on source separation or transportation of sorted recyclables. When the number of people living in the same household is large,

naturally they generate more waste and may need more space to temporarily store them if frequent collection mechanisms are not facilitated. Since this survey population is a good representation of the majority who live in Zeytinburnu Municipality, the future success of their recycling program depends on how effectively these difficulties can be addressed. Introducing more collection centers and/or more frequent collections could be something to consider, although the same may cost more money to maintain the program. As mentioned before, making recycling not only affordable, but also approachable is an effective way to enhance its adoption [14,15]. Perhaps, teaming up with other organizations such as community centers and/or supermarkets/grocery stores and potentially using such places as alternative collection centers may offer a cost-effective solution to expand the collection network. In particular, facilitating more collection points at public places (such as at established supermarkets) can positively influence the social norms associated with recycling through high visibility. Therefore, addressing these issues are important, not only because it can improve this particular recycling program in Zeytinburnu, but also due to the long-lasting positive behavioral trends it may set towards a pro-recycling culture.

## 7. Conclusions

The main conclusions of this research can be briefly summarized as follows: About 85% of the survey takers learned about the recycling program through such methods, while only 1% learned about it through traditional means such as billboards and governmental gazettes. This suggests that visual observations and word of mouth can be more effective channels in spreading the news about the recycling program in an urban setting. The major challenges that the survey takers faced during their participation in the recycling program were more related to a lack of infrastructure than a lack of will. Difficulties in storing waste at home (32%) and transporting it to the collection centers (46%) were the major challenges faced by the Zeytinburnu recycling program participants. Interestingly, a lack of time (8%) or a lack of knowledge on source separation (3.5%) were found to be minor issues.

Recycling consciousness of the survey takers seemed to be positively correlated with their knowledge/exposures, as observed through indicators such as education and/or profession. More than half of them believed that recycling programs can be made more active and effective with better education/instructions. Such educational activities may also help the population realize that they should also take part of the recycling responsibility. About half of the survey respondents admitted that they were in the recycling program for financial incentives and would not continue when/if the incentives are not offered. For the program to be sustainable in the long run, it is essential to make this half more recycling conscious. The data presented support that this may also be achieved through educational- or awareness-raising programs.

**Author Contributions:** Study conceptualization, survey design, data collection, preliminary analysis, and the preparation of the first draft were conducted by B.H.G.H. and A.A.; Review and secondary analysis by H.H.; Final draft of the manuscript was written by B.H.G.H. and H.H. All authors have read and agreed to the published version of the manuscript.

**Funding:** The research was partly funded by The Scientific and Technological Research Council of Turkey (with the number 1919B012105189).

**Institutional Review Board Statement:** The study was conducted in accordance with the Declaration of Marmara University and approved by the Marmara University Social Science Institute Ethics Committee (protocol code: 2022-3/14; date: 16 May 2022).

**Informed Consent Statement:** Informed consent was obtained from all respondents involved in the study.

**Data Availability Statement:** The authors confirm that the data supporting the findings of this study are available within the article.

**Acknowledgments:** This study was carried out by the Urban Problems and Local Government Research and Implementation Centre, Marmara University, and Zeytinburnu Municipality, with the

project supported by the Scientific and Technological Research Council of Turkey (with the number 1919B012105189). The authors would like to thank to Beyza Takta Varli and Ilknur Celik, and the Zeytinburnu Municipality staff for the support they receive during the study.

**Conflicts of Interest:** The authors declare no conflict of interest.

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
