# Peer review of "The Social Dimensions of an Incentive-Based Urban Recycling Program: A Case-Study from Istanbul, Turkey"

_sustainability, doi:10.3390/su152215775_

Round 1
Reviewer 1 Report
Comments and Suggestions for Authors
The authors use a self-administered survey to determine the recycling behavior/knowledge in Zeytinburnu Municipality in Istanbul (Turkey). Their sample includes 428 respondents. The results revealed that about half of the respondents are spontaneous recyclers and the other half supported the program just for the reward.
The topic, methodology, and used data are up to date. The selection of the study area is relevant, as Zeytinburnu is a district/municipality in the greater Istanbul area, which is the most populous city in Turkey and an important cultural and economic center. Zeytinburnu, however, seems to be a rather poor community (in terms of income and education) within Turkey, which makes it an even more interesting case study for the topic of recycling.
To my knowledge, there was not published a comparable study in English language about this topic related to the area of study. The used methods are adequate, and the illustrative figures are useful. Overall, the article is written well and represents high scientific standards. I really enjoyed reading it.
Overall, I recommend accepting the manuscript, after some minor revisions are done. Here are some comments that might help to improve the quality of the manuscript:
1) It will be helpful if the introduction mentions the sample size of the survey and mentions the year when the survey took place.
2) Lines 189-191: How was the minimum number of the sample size determined? Maybe you can add 1-2 sentences. Did you use Slovin’s formula? I get slightly different results, when I use it.
3) In the research methodology part, you can maybe add information if the survey was conducted by an interviewer or did the participants fill a sheet of paper alone. Was the concept of recycling explained? Did respondents know what exactly it means? Maybe you can add a couple of sentences about that.
4) Table 3 shows the results for two different groups. However, I do not see any statistical tests that show that the differences are statistically significant. I suggest you to use at least a t-test or ANOVA test, which can be used to compare two samples. If you do not show that the difference is statistically different from zero on certain significance levels, you will not have a statistically meaningful difference.
5) The collected dataset can be explored more, maybe not for this paper, but I see potential to use the six main questions as outcome variables and then use multinominal logit/probit or binary logit/probit to understand more about the role of respondents’ characteristics in answering these questions. There could also be interesting relationships between the six questions themselves, so these should also be added as control variables. Again, this will exceed this paper, and is just a recommendation for further research.
Author Response
Kindly refer to the attached pdf.

Reviewer 2 Report
Comments and Suggestions for Authors
The manuscript is well-written, methodologically sound, and presents interesting results of considerable relevance to policymakers in particular. Furthermore, it aligns well with the scope of Sustainability.
However, there are some – mostly minor to modearate - issues that need to be addressed before the manuscript can be considered for publication.
*Major
The abstract should be revised to specify the sample size (N), detail the sampling procedure used, and describe the mode of the survey (face-to-face, online…) conducted.
Introduction:
The research is relatively well-grounded in existing literature, although litearture on social influence is largely absent. Also, the research question lacks clarity and could be better framed. A notable concern is that once financial incentives are withdrawn, the desired behavior often declines, sometimes even below levels seen before the incentives were introduced (motivation crowding out, refer to Bowles and Reyes 2012). This phenomenon is frequently addressed in the context of PES (payment for ecosystem services), as highlighted by Vorlaufer (2023) and Maca Millan (2021). Social factors can promote the desired behavior as a lasting social norm after incentives are removed, especially if the incentives lead to widespread adoption of the behavior. See Berger (2023) for a theoretical framework for a practical application related to reuse."
Bowles, S., & Polania-Reyes, S. (2012). Economic incentives and social preferences: substitutes or complements?. Journal of Economic Literature, 50(2), 368-425.
Berger, J., Efferson, C., & Vogt, S. (2023). Tipping pro-environmental norm diffusion at scale: opportunities and limitations. Behavioural public policy, 7(3), 581-606.
Maca-Millan, S., Arias-Arevalo, P., & Restrepo-Plaza, L. (2021). Payment for ecosystem services and motivational crowding: experimental insights regarding the integration of plural values via non-monetary incentives. Ecosystem Services, 52, 101375.
Vorlaufer, T., Engel, S., de Laat, J., & Vollan, B. (2023). Payments for ecosystem services did not crowd out pro-environmental behavior: Long-term experimental evidence from Uganda. Proceedings of the National Academy of Sciences, 120(18), e2215465120.
Theory/Literature
The authors correctly note that attitudes don't always lead to action. While information is crucial, motivation to act is equally important. It could be beneficial to mention that making recycling less costly or effortful can promote its adoption. When the gap between public and private benefits narrows, the desired behavior becomes more probable (refer to Diekmann 2004, Berger 2023). This idea aligns well with the policy recommendation on p. 12.
Diekmann, A., & Preisendörfer, P. (2003). Green and greenback: The behavioral effects of environmental attitudes in low-cost and high-cost situations. Rationality and Society, 15(4), 441-472.
Methods
High percentage of women in the sample: Once could also argue that the study addresses households, not individuals. As such, the data is still informative regarding recycling behavior of households. The authors could, for example, present data on the percentage of the population living in single households. Likely, these individuals (male single, in particular) are underrepresented in the sample. So the authors could get an idea of the bias in their sample (which is presumably small).
Results
The results are informative, yet, they could be presented in a more accessible way. A graph (e.g. bars, preferably including confidence intervals) might be a great way to present this kind of data.
Discussion
The authors could delve deeper into the concept of the social tension between private and public benefits or costs associated with recycling. The policy recommendations are logical when viewed through this lens. Reducing the costs and amplifying the benefits of recycling both drive its adoption. As recycling becomes a societal norm, benefits arise from social acknowledgment. Early adopters can earn reputational advantages by showcasing their readiness to bear costs or exert effort for the collective benefit (refer to Brick 2017, Berger 2019). The further spread of the target behavior could be promoted by means of information on social norms (normative appeals, descriptive feedback; see Goldstein et al. 2008).
Brick, C., Sherman, D. K., & Kim, H. S. (2017). “Green to be seen” and “brown to keep down”: Visibility moderates the effect of identity on pro-environmental behavior. Journal of Environmental Psychology, 51, 226-238.
Berger, J. (2019). Signaling can increase consumers' willingness to pay for green products. Theoretical model and experimental evidence. Journal of consumer behaviour, 18(3), 233-246.
Goldstein, N. J., Cialdini, R. B., & Griskevicius, V. (2008). A room with a viewpoint: Using social norms to motivate environmental conservation in hotels. Journal of consumer Research, 35(3), 472-482.

The manuscript is well-written, methodologically sound, and presents interesting results of considerable relevance to policymakers in particular. Furthermore, it aligns well with the scope of Sustainability.
However, there are some – mostly minor to modearate - issues that need to be addressed before the manuscript can be considered for publication.
*Major
The abstract should be revised to specify the sample size (N), detail the sampling procedure used, and describe the mode of the survey (face-to-face, online…) conducted.
Introduction:
The research is relatively well-grounded in existing literature, although litearture on social influence is largely absent. Also, the research question lacks clarity and could be better framed. A notable concern is that once financial incentives are withdrawn, the desired behavior often declines, sometimes even below levels seen before the incentives were introduced (motivation crowding out, refer to Bowles and Reyes 2012). This phenomenon is frequently addressed in the context of PES (payment for ecosystem services), as highlighted by Vorlaufer (2023) and Maca Millan (2021). Social factors can promote the desired behavior as a lasting social norm after incentives are removed, especially if the incentives lead to widespread adoption of the behavior. See Berger (2023) for a theoretical framework.
Bowles, S., & Polania-Reyes, S. (2012). Economic incentives and social preferences: substitutes or complements?. Journal of Economic Literature, 50(2), 368-425.
Berger, J., Efferson, C., & Vogt, S. (2023). Tipping pro-environmental norm diffusion at scale: opportunities and limitations. Behavioural public policy, 7(3), 581-606.
Maca-Millan, S., Arias-Arevalo, P., & Restrepo-Plaza, L. (2021). Payment for ecosystem services and motivational crowding: experimental insights regarding the integration of plural values via non-monetary incentives. Ecosystem Services, 52, 101375.
Vorlaufer, T., Engel, S., de Laat, J., & Vollan, B. (2023). Payments for ecosystem services did not crowd out pro-environmental behavior: Long-term experimental evidence from Uganda. Proceedings of the National Academy of Sciences, 120(18), e2215465120.
Theory/Literature
The authors correctly note that attitudes don't always lead to action. While information is crucial, motivation to act is equally important. It could be beneficial to mention that making recycling less costly or effortful can promote its adoption. When the gap between public and private benefits narrows, the desired behavior becomes more probable (refer to Diekmann 2004, Berger 2023). This idea aligns well with the policy recommendation on p. 12.
Diekmann, A., & Preisendörfer, P. (2003). Green and greenback: The behavioral effects of environmental attitudes in low-cost and high-cost situations. Rationality and Society, 15(4), 441-472.
Methods
High percentage of women in the sample: Once could also argue that the study addresses households, not individuals. As such, the data is still informative regarding recycling behavior of households. The authors could, for example, present data on the percentage of the population living in single households. Likely, these individuals (male single, in particular) are underrepresented in the sample. So the authors could get an idea of the bias in their sample (which is presumably small).
Results
The results are informative, yet, they could be presented in a more accessible way. A graph (e.g. bars, preferably including confidence intervals) might be a great way to present this kind of data.
Discussion
The authors could delve deeper into the concept of the social tension between private and public benefits or costs associated with recycling. The policy recommendations are logical when viewed through this lens. Reducing the costs and amplifying the benefits of recycling both drive its adoption. As recycling becomes a societal norm, benefits arise from social acknowledgment. Early adopters can earn reputational advantages by showcasing their readiness to bear costs or exert effort for the collective benefit (refer to Brick 2017, Berger 2019). The further spread of the target behavior could be promoted by means of information on social norms (normative appeals, descriptive feedback; see, Goldstein et al. 2008).
Brick, C., Sherman, D. K., & Kim, H. S. (2017). “Green to be seen” and “brown to keep down”: Visibility moderates the effect of identity on pro-environmental behavior. Journal of Environmental Psychology, 51, 226-238.
Berger, J. (2019). Signaling can increase consumers' willingness to pay for green products. Theoretical model and experimental evidence. Journal of consumer behaviour, 18(3), 233-246.
Goldstein, N. J., Cialdini, R. B., & Griskevicius, V. (2008). A room with a viewpoint: Using social norms to motivate environmental conservation in hotels. Journal of consumer Research, 35(3), 472-482.
Author Response
Kindly refer to the attached pdf.

Round 2
Reviewer 2 Report
Comments and Suggestions for Authors
The authors have thoroughly revised their manuscript and there are only a few minor points left to address before publication. Some of them, however, are rather severe, so I recommend minor revisions. But let me clear: The revision of Figures 4 and 5, ensuring they are presented in a meaningful format, is a prerequisite for publication.
Comments on the Quality of English LanguageThe authors have thoroughly revised their manuscript and there are only a few minor points left to address before publication. Some of them, however, are rather severe, so I recommend minor revisions. But let me clear: The revision of Figures 4 and 5, ensuring they are presented in a meaningful format, is a prerequisite for publication.
On p. 7 the authors claim: “The number of responses received for each answer and the same reported as percentages are also presented in Table 2 (Columns 2 and 3, respectively) and the confidence intervals are displayed 253 in Figure 4.”
However, Figure 4 does not display any confidence intervals. Since box plots do not depict confidence intervals, the authors should either introduce a figure that includes them or remove the assertion while retaining the box plots.
Furthermore, box plots are appropriate for metric variables and not for categorical variables. Thus, Figure 4 may not effectively represent the data. Consider Q1: "Which of the following do you think is non-recyclable waste?". A more meaningful way to present the results would be through a bar chart that shows the percentage of respondents correctly identifying each item as non-recyclable waste. While it would be ideal for the bar charts to feature confidence intervals, the decision to incorporate them is at the discretion of the authors.
Also, Figure 5 is erroneous. Gender, when measured as a binary variable, should be represented, for example, with a bar or pie chart. Please revise Figures 4 and 5.
Minor
On p. 3, the word “incentives” is spelled incorrectly (missing “-“): Although the financial incenti ves are thought to make a positive effect on waste recycling behaviors in general [28, 32], 114 a few studies have expressed doubts about their effectiveness [33, 34, 35].
Author Response
Please see the attached pdf for the explanation on how we improved our paper based on your comments.
Another mistake we found in Table 1 was also corrected in this version. While formatting the tables as per MDPI requirements, we have inadvertently missed the last item in Table 1, i.e., Household Population data (which was already explained within the text as well as in Table 3). We revised Table 1 to re-include this missing information and it is highlighted in yellow for your easy reference.
Many thanks for your help.
